# Effectiveness and cost-effectiveness of implementing HIV testing in primary care in East London: protocol for an interrupted time series analysis

Werner Leber,[1] Lee Beresford,[1] Claire Nightingale,[2] Estela Capelas Barbosa,[3] Stephen Morris,[3] Farah El-Shogri,[1] Heather McMullen,[1] Kambiz Boomla,[1] Valerie Delpech,[4] Alison Brown,[4] Jane Hutchinson,[5] Vanessa Apea,[5] Merle Symonds,[5] Samantha Gilliham,[5] Sarah Creighton,[6] Maryam Shahmanesh,[3] Naomi Fulop,[3] Claudia Estcourt,[5,7] Jane Anderson,[6] Jose Figueroa,[8] Chris Griffiths[1]

For numbered affiliations see end of article.

**Correspondence to**
Dr Werner Leber;
w.leber@qmul.ac.uk

## ABSTRACT

**Introduction** HIV remains underdiagnosed. Guidelines recommend routine HIV testing in primary care, but evidence on implementing testing is lacking. In a previous study, the Rapid HIV Assessment 2 (RHIVA2) cluster randomised controlled trial, we showed that providing training and rapid point-of-care HIV testing at general practice registration (RHIVA2 intervention) in Hackney led to cost-effective, increased and earlier diagnosis of HIV. However, interventions effective in a trial context may be less so when implemented in routine practice. We describe the protocol for an MRC phase IV implementation programme, evaluating the impact of rolling out the RHIVA2 intervention in a post-trial setting. We will use a longitudinal study to examine if the post-trial implementation in Hackney practices is effective and cost-effective, and a cross-sectional study to compare Hackney with two adjacent boroughs providing usual primary care (Newham) and an enhanced service promoting HIV testing in primary care (Tower Hamlets).

**Methods and analysis** Service evaluation using interrupted time series and cost-effectiveness analyses. We will include all general practices in three contiguous high HIV prevalence East London boroughs. All adults aged 16 and above registered with the practices will be included. The interventions to be examined are: a post-trial RHIVA2 implementation programme (including practice-based education and training, external quality assurance, incentive payments for rapid HIV testing and incorporation of rapid HIV testing in the sexual health Local Enhanced Service) in Hackney; the general practice sexual health Network Improved Service in Tower Hamlets and usual care in Newham. Coprimary outcomes are rates of HIV testing and new HIV diagnoses.

**Ethics and dissemination** The chair of the Camden and Islington NHS Research Ethics Committee, London, has endorsed this programme as an evaluation of routine care. Study results will be published in peer-reviewed journals and reported to commissioners.

## Strengths and limitations of this study

► The study will use existing trial data preceding the implementation programme in addition to post-trial data, collected using similar methods to those used in the trial.

► Fully anonymised general practice data from all participating boroughs will be remotely extracted using a single general practice computer system (Egton Medical Information Systems, EMIS).

► A large dataset with at least 12 time points before and at least 29 time points after the intervention implementation will provide precise estimation of the effects of intervention using interrupted time series analysis.

► Differences in HIV testing protocols, data quality and extraction between Hackney, and Tower Hamlets and Newham; and previous interventions such as rapid HIV testing pilots in a small number of Newham practices, may affect comparability between boroughs. Also, cross-contamination of Rapid HIV Assessment 2 (RHIVA2) control practices from intervention practices, or between boroughs, may have occurred.

► Finally, the post-trial implementation programme of the RHIVA2 intervention differed slightly from the research intervention to account for the complexities of delivering services in routine healthcare settings.

## INTRODUCTION

HIV prevalence is increasing, with over 100 000 people now estimated to be living with HIV in the UK.[1] From 1999 to 2013, the numbers of patients accessing specialist care increased fourfold.[2 3] Successfully treated, HIV now has characteristics of a long-term condition, with patients taking life-long treatment and increasingly exhibiting age-related comorbidities, such as cardiovascular disease, osteoporosis and mental illness. The cost of HIV care for the National Health Service (NHS) (without considering the cost of treating comorbidities) in 2013

was estimated to be £750 million (£1 billion with social care included).[4]

Late diagnosis of HIV increases morbidity and mortality, and is associated with unwitting onward transmission. To reduce the pool of undiagnosed infection in the population, the British HIV Association (BHIVA) and National Institute for Health and Care Excellence (NICE) recommended expansion of routine HIV testing from sexual health and antenatal clinics to non-traditional settings including general practices located in high HIV prevalence areas (two or more people diagnosed with HIV/1000 adult population).[5–8] In the Framework for Sexual Health Improvement in England (2013), the UK Department of Health additionally set out recommendations for the commissioning of HIV testing in high prevalence areas across all healthcare settings, including general practice.[9] However, the decision to implement these recommendations lies with the 74 local authorities in England and the respective clinical commissioning groups (CCGs). Although National Policy for HIV screening in antenatal settings has proven highly effective, guidance for screening in general practice is lacking. Indeed, it was hoped that recommending routine HIV testing would suffice to enhance awareness of HIV among general practitioners (GPs) and become standard in this setting.[10] Pilot studies have demonstrated feasibility and acceptability of routine HIV testing among patients and healthcare professionals.[11 12] Despite this, a recent systematic review identified low uptake of routine HIV testing in non-traditional UK settings including general practice following the BHIVA 2008 guidelines.[13]

In a large-scale pragmatic cluster randomised controlled trial (Rapid HIV Assessment 2 (RHIVA2)) in the London borough of Hackney, we showed that rapid HIV testing offered at general practice registration was effective and safe, and delivered increased and earlier HIV diagnosis.[14] Furthermore, a recent health economics analysis of the RHIVA2 trial demonstrated that HIV screening in primary care in high HIV prevalence areas is cost-effective.[15] Following completion of this trial, we implemented HIV testing into routine care across all Hackney practices using a theory-based programme[16–18] comprising an adaptation of the RHIVA2 intervention, including a practice-based education programme, a borough-wide audit of HIV care, and the introduction of payments for routine HIV testing as part of an updated general practice sexual health Local Enhanced Service (LES).[19]

We will evaluate both the RHIVA2 trial and post-trial implementation programme through an MRC phase IV implementation study.[20] The study will also compare the impact of RHIVA2 implementation in Hackney practices with two neighbouring boroughs; Tower Hamlets that offers a general practice sexual health Network Improved Service (NIS) promoting routine HIV testing; and Newham offering usual primary care without incentivised HIV testing.

Figure 1 summarises the timelines of HIV-related interventions in these three boroughs.

The study will therefore address the following questions:

- ► How effective and cost-effective is the RHIVA2 intervention when implemented as a clinical service outside a research context?
- ► How do effectiveness and cost-effectiveness of the RHIVA2 trial and implementation programme compare with a borough-wide sexual health NIS promoting routine HIV testing in Tower Hamlets?
- ► How do both the RHIVA2 trial and post-trial implementation programme and the Tower Hamlets sexual health NIS compare with Newham that offers usual HIV primary care only?

## METHODS AND ANALYSIS
### Setting
The study will be set in the three East London boroughs of Hackney, Tower Hamlets and Newham. The estimated diagnosed HIV prevalence for these boroughs is 8.1, 6.5 and 6.7 per 1000 adult population, respectively.[21] We will include all adults aged 16 and above registered with general practices in the three boroughs.

### Design
Service evaluation comprising a pragmatic cohort using an interrupted time series (ITS) analysis and cost-effectiveness analysis. The evaluation will examine the longitudinal impact of interventions in Hackney and the comparative cross-sectional impact of interventions in Hackney, Tower Hamlets and Newham. HIV testing and diagnosis data from 1 April 2009 to 30 June 2015 is being collected.

### Interventions
#### Hackney
The RHIVA2 intervention was first introduced in April 2010 in 20 general practices randomised to the intervention arm of the RHIVA2 trial (table 1). This theory-based intervention[16–18] is described in detail in Leber *et al*[14] and consisted of a practice-based educational training session for the primary care team to promote rapid HIV testing at registration, a follow-up meeting with a nominated practice lead nurse, integration of rapid HIV testing with the general practice computer template, and an incentive payment of £10 per rapid HIV test performed. Testing competence was monitored by an external quality assessment. This intervention was offered in addition to an existing national antenatal HIV screening service offered at the practices, and a sexual health LES promoting sexual health screening including HIV case detection (incentive payments of £265 per newly diagnosed patient) introduced in 2006/2007.[19]

The RHIVA2 post-trial implementation programme ran from September 2012 to June 2015 and comprised the following three components[16–18]:

1. Additional post-trial HIV testing training (September 2012 to February 2013). This was offered to all Hackney practices at the completion of the RHIVA2 trial and comprised a modification of the RHIVA2 trial intervention whereby practices were encouraged to

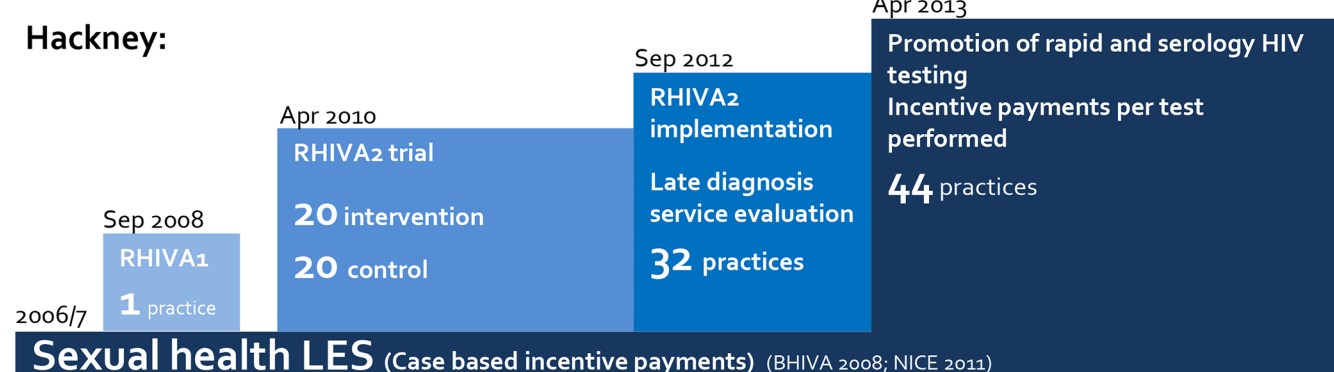

**Figure 1** Sexual health service provision in East London general practice. In Hackney (blue), the Rapid HIV Assessment (RHIVA) research programme promoting rapid point-of-care HIV testing was developed in addition to a sexual health local enhanced service (LES) and included: a pilot study (RHIVA1), a cluster randomised controlled trial across 20 intervention practices (RHIVA2) and implementation of the RHIVA2 intervention across all Hackney practices. In April 2013, the RHIVA2 intervention was integrated with the sexual health LES. In Tower Hamlets (red), a sexual health network improved service (NIS) replaced the previous LES in April 2011. Newham (green) does not provide a service promoting HIV testing (usual care). BHIVA, British HIV Association; DoH, Department of Health; NICE, National Institute for Health and Care Excellence.

offer both routine serology and rapid HIV testing in any clinical setting (rather than just rapid testing at practice registration).

2. Updated sexual health LES. From April 2013, the existing LES was replaced with HIV testing payments to practices of between £7 and £10 for every rapid or serology HIV test carried out in any clinical setting, in addition to a payment of £258 for any new diagnosis.

3. General practice-based HIV care audit (October 2012 to March 2013) of missed diagnoses of HIV and safety of coprescribing with antiretroviral medication in Hackney general practices.[22]

### Tower Hamlets

In April 2011, Tower Hamlets implemented sexual health NIS incentive payments promoting cooperation between practices in eight practice networks, comprising four to five practices each.

The NIS incentivised sexually transmitted infections (STIs) testing with separate payments for serology (£10 per HIV test combined with or without hepatitis B and syphilis testing) and swabs/urine (£15 for gonorrhea and chlamydia testing).

Practices were encouraged to offer HIV and STI testing as part of the new patient check as well as in sexual health and contraception consultations and opportunistically in general consultations.

In contrast to Hackney, where practices received the implementation individually over a 6-month period, the Tower Hamlets sexual health NIS went live through activation of the computerised sexual health template on the same day for all practices (April 2011). A primary care facilitator in sexual health (JH since September 2012) has additionally provided regular support, training and performance feedback to the practice networks.

### Newham

*Usual care*—no specific borough-wide initiatives were promoted relating to HIV testing in primary care.

The RHIVA2 intervention is based on published clinician behaviour change programmes,[16–18] and the Tower Hamlets NIS follows Royal College of General Practitioners guidance on establishing general practice federations.[23] We will report our study in accordance with the STARI reporting guidelines for implementation studies.[24]

### Outcome measures

All data gathered will comprise routine clinical data. No identifiable data will be held by the research team.

**Table 1** Schedule for enrolment, interventions and assessment of the study

| Timepoint | Control period | | | Practice allocation | Post-allocation | | | Close-out | | |
|---|---|---|---|---|---|---|---|---|---|---|
| | −41 months | −24 months | −12 months | 0 months | +1 months | +2 months | ≥12 months | +29 months | +34 months | +51 months |
| RHIVA2 trial: | | | X | | | | | | | |
| Allocation | | | | X | | | | | | |
| Intervention | | | | | X | | | | | |
| Enrollment | | | | | X | X | X | | | |
| RHIVA2 post-trial implementation: | X | | | | | | | | | |
| Allocation | | | | X | | | | | | |
| Intervention | | | | | X | | | | | |
| Enrollment | | | | | X | X | X | X | | |
| Tower Hamlets: | | X | | | | | | | | |
| Allocation | | | | X | | | | | | |
| Intervention | | | | X | | | | | | |
| Enrollment | | | | X | X | X | X | X | X | X |
| Assessments: | | | | | | | | | | |
| HIV testing | X | X | X | X | X | X | X | X | X | X |
| HIV diagnosis | X | X | X | X | X | X | X | X | X | X |

RHIVA2, Rapid HIV Assessment 2.

## Coprimary outcome measures

► Rates of HIV testing (combined rapid and serology testing; number of patients tested/1000 population/year)
► Rates of new HIV diagnosis (number of newly diagnosed patients/10 000/year)

## Secondary outcome measures

Secondary outcome measures will include data on type of HIV test (serology; rapid), location of diagnosis, stage of disease and linkage to secondary care and retention in secondary care:

► Number/rate of patients who received a rapid HIV test.
► Number/rate of patients who received a serology HIV test.
► Number/rate of patients newly diagnosed in general practice.
► Mean and median CD4 count of patients newly diagnosed in primary care.
► Proportion of patients with a CD4 count <350 cells/µL of blood.
► Proportion of patients with a CD4 count <200 cells/µL of blood.
► Proportion of patients newly diagnosed with HIV in primary care who attend an HIV specialist department within 1 week of diagnosis.
► Proportion of patients newly diagnosed with HIV in primary care who attend an HIV specialist department within 2 weeks of diagnosis (BHIVA Standards of Care for people living with HIV (PLHIV), Standard 2, 2013).[25]
► Proportion of patients newly diagnosed with HIV in primary care who attend an HIV specialist department within 4 weeks of diagnosis (NHS England Standard).
► Proportion of patients newly diagnosed with HIV in primary care who attend a HIV specialist department within 3 months of diagnosis (Optimising testing and linkage to care for HIV across Europe, OptTEST, 2013).
► Number of patients previously known to have HIV, who attend an HIV specialist clinic between 12 and 24 months following new diagnosis in primary care (BHIVA Standards of Care for PLHIV, Standard 2, 2013).

## Data sources

We will gather data from April 2009 to June 2015 using the following sources.

### Organisational data

For Hackney practices, we hold detailed records of RHIVA2 trial educational training sessions, post-trial education sessions and participation in the borough-wide audit of HIV care. For Tower Hamlets, we hold data on timing of the introduction of the NIS. For all boroughs, we will seek to obtain any other relevant activities promoting HIV testing in the practices.

## Clinical data
### HIV diagnosis data

Borough-specific master case report forms (CRF) containing lists of patients with a positive confirmatory HIV test result will be generated by the lead HIV clinician from the clinical HIV services in each borough. This master CRF will be shared with the study team in an anonymised format as per the RHIVA2 protocol. At Homerton Hospital, Hackney data on newly diagnosed patients have been collected prospectively as outlined in RHIVA2 (continuous data available from 2009). For Tower Hamlets and Newham, new diagnosis data will be collected retrospectively for the same period. Clinical leads will liaise with Public Health England to categorise patients with a positive confirmatory HIV test into newly diagnosed patients and people already known to be HIV positive, and obtain data on patients who had defaulted from care between 12 and 24 months following diagnosis.

### HIV testing data

In collaboration with the Clinical Effectiveness Group in the Centre for Primary Care and Public Health (QMUL), we will collect anonymised data on HIV testing in primary care, as per RHIVA2 protocol, using remote searches of borough-specific READ codes used for rapid and serology HIV testing.

## Sample size and analysis
### Sample size

Sample size calculations for the ITS analysis are based on the number of time points available to perform the analysis. Generally, a minimum of 10–12 time points before and after an intervention are required to determine statistical significance of an intervention.[26 27] Twelve monthly time points are available before the RHIVA2 trial intervention and a minimum of 13 time points for the period thereafter; and 41 monthly time points are available before the RHIVA2 implementation and a minimum of 29 time points thereafter. For Tower Hamlets, 24 monthly time points are available before the introduction of the sexual health NIS and a minimum of 51 time points thereafter.

### Longitudinal analysis of effectiveness of RHIVA2 implementation programme in Hackney

We will compare testing rates using an ITS regression approach with a mixed Poisson model (or negative binomial model if there is overdispersion) to analyse the effect of interventions. Mixed Poisson regression will allow for the clustered nature of the data, that is, allowing each practice to have a different intercept (baseline HIV testing rate). We will also adjust for practice list size as an offset variable in the analysis. Using this approach, it will be possible to examine whether there is a change in level and/or a change in regression slope following the implementation of the intervention.

We will first examine the impact of the RHIVA2 intervention on combined serology and rapid HIV testing rate in practices that received the RHIVA2 implementation (post-trial setting, comparing testing rates preintervention and postintervention). We will repeat this analysis including practices which did not receive the RHIVA2 intervention during the trial or the further implementation phase, to act as a comparator. This will allow comparison of testing rates in practices postimplementation to their preimplementation rates, also adjusting for continuing trends in practices that did not receive the RHIVA2 implementation. We will also repeat the analysis using Newham practices (which received no intervention) as a further comparator.

The date of intervention will be defined as the date of the first RHIVA2 training session. We will conduct similar analysis to the above, with the endpoint of HIV diagnosis, rather than testing.

To assess sustainability, we will model testing rates in practices that received the RHIVA2 intervention during the trial period, considering an interaction term between time postintervention and whether practices were retrained. This will allow an assessment to be made about whether retraining influences the slope postintervention, that is, the sustainability of the RHIVA2 intervention.

### Cross-sectional analysis of effectiveness of the RHIVA2 implementation programme in Hackney, a sexual health NIS in Tower Hamlets and usual care in Newham

We will examine the impact of the sexual health NIS on the serology testing rate in all Tower Hamlets practices (comparing testing rates preimplementation and postimplementation). We will repeat this analysis but also including practices in Newham, which received no intervention at any stage, to act as a comparator. The date of intervention will be defined as the date when the sexual health NIS system went live for all practices (April 2011). We will also compare the RHIVA2 implementation in Hackney with the introduction of the sexual health NIS in Tower Hamlets.

We will conduct similar analysis to the above, with the endpoint of HIV diagnosis, rather than testing.

### Cost-effectiveness analysis

Cost-effectiveness analyses for the longitudinal and cross-sectional analyses described above will build on a cost-effectiveness model developed for the RHIVA2 trial.[15] To assess value for money, we will identify the difference in costs and the difference in benefits of implementing the RHIVA2 intervention and we aim to distinguish between two different analyses. First, we will evaluate the cost-effectiveness of the RHIVA2 implementation using the same 'comparator' defined in the longitudinal analysis, that is, practices which did not receive the RHIVA2 intervention at any stage will be considered the 'absence of the intervention'.

Second, we will build a cost-effectiveness comparison between different forms of HIV screening in East London, formally comparing the RHIVA2 implementation (conducted in Hackney) with the sexual health

NIS (Tower Hamlets) and usual care (Newham). Once again, all three different approaches will be compared with practices, which did not receive the RHIVA2 intervention at any stage, as these are considered the 'absence of the intervention'. Given that the same control group of practices in Hackney will serve as reference, this second analysis will formally compare the different approaches from different boroughs, considering no intervention in Hackney as a reference.

Three main cost categories will be explored: (1) the extra start-up cost of RHIVA2 implementation (specifically the cost for RHIVA2 staff time, including follow-up activities; cost of training; cost of training materials and cost of incentives to practices); (2) the extra recurrent cost for performing rapid HIV testing and (3) additional costs associated with reactive rapid HIV test results.

Benefits of the RHIVA2 programme will include health outcomes for patients, specifically deaths averted, secondary HIV infections averted and quality-adjusted life years; these are a function of the uptake of screening and screening positivity, which will be quantified directly by the analyses.

This study is a service evaluation which does not require registration as a trial.

## ETHICAL ASPECTS

The programme described is an evaluation of routine care. Its status as such has been endorsed by the chair of the Camden and Islington NHS Research Ethics Committee London. The committee's view is that the work does not require NHS ethics approval. The programme will be implemented in partnership with the local providers and commissioners and involve the analysis of existing routine data. This also applies to data on delayed diagnosis of HIV (CD4 count), and demographics reported in the RHIVA2 trial, which is routinely collected as part of a national audit (BHIVA National Audit 2010) and has been included in Public Health Outcomes Framework (http://www.phoutcomes.info). We will inform the research ethics committee, the CCG and Public Health about any important changes to the protocol.

This service evaluation is to answer the question: what standard does this service achieve? It involves routine implementation of an intervention outside of the research context. The actual management options are those of the clinician and patient only. There is no random allocation to intervention and control groups. No individual consent is required from newly diagnosed patients.

In addition to CD4 count and viral load, the following additional data will be obtained by the clinical leads of specialist departments in an anonymised fashion and stored in accordance with good clinical practice:
- ► Most likely mode of HIV acquisition
- ► Sexual orientation
- ► Soundex code
- ► Ethnicity
- ► Gender
- ► Age at diagnosis.

## DISCUSSION

The MRC Framework for complex interventions[28] clearly describes the need for implementation studies that test the effectiveness of interventions found to be promising in phase III randomised controlled trials. Successful translation of such interventions into routine care cannot be assumed; trials take place in tightly controlled conditions that aim to limit bias, with selected (often motivated) participants, while implementation programmes (usually) aim much more broadly to include all potential participants.[29] Trial interventions will often need to be augmented by a distinct implementation package. Their success will be affected by multiple factors including, but not limited to contextual factors such as, differences in setting, population, organisation of healthcare systems, competing initiatives and financial aspects (including availability of incentives). Evaluations of implementation of trials into routine care are frequently not carried out, and if they are, are often poorly reported.[30] Such evaluations are highly unlikely to involve randomised designs, and are more likely to be observational, including the use of service evaluations. Designs are often constrained by health systems and data availability, but need to be as robust as possible. New reporting guidelines have recently been published that highlight the need for evaluation of implementation and aim to improve their reporting, such that they can be more easily accessed in the literature and serve to guide others in designing the most effective implementation programmes.[29]

Recent NICE guidelines recommend expanded HIV testing in primary care,[6–8] but little information is available about how to operationalise testing, especially in the context of current resource constraints in primary care. It is, therefore, vital that innovative and effective ways are found to implement testing. The strengths of the proposed work include the availability of existing trial data preceding the implementation programme, the capacity to collect data longitudinally using similar methods to those used in the trial, and the use by general practices across all three participating boroughs of a single GP computer system supplier Egton Medical Information Systems, data from which can be remotely and electronically (with full anonymisation) searched and collated by the research team (http://www.blizard.qmul.ac.uk/ceg-home.html), confirmation of HIV test results for all study practices at the same tertiary care laboratory (Barts Health NHS Trust) and external data validation by Public Health England. This approach provides a unique capability, allowing comparative data on three separate approaches to HIV testing and diagnosis to be accurately gathered and analysed.

Due to difference in data extraction between Hackney and Tower Hamlets and Newham, respectively, there may be a small chance of differential reporting bias between these boroughs. In contrast to the Department of Sexual

Health at Homerton University Hospital (Hackney), which is able to track any patient lost to follow up using a monthly list of HIV positives provided by the laboratory, no such fail-safe procedure exists in Tower Hamlets or Newham. However, in these latter boroughs all data for HIV positive test results are available on the general practice computer system via an electronic laboratory link, almost eliminating the chance of missing data. We are currently conducting an audit of outcomes from people newly diagnosed with HIV in Tower Hamlets and Newham general practices to extract this data for the study period specified.

We cannot exclude occurrence of any cluster or cross-over effects, although we would expect for this to be minimal given that we detected a more than fourfold increase in HIV testing rates between the RHIVA2 trial intervention and control practices. In contrast to the network structure in Tower Hamlets, Hackney practices, which were randomly allocated to the trial arms, may have informally worked collaboratively through geographical proximity. Due to this potential clustering effect, inference of generalisability of RHIVA2 findings in other boroughs should, therefore, be treated with caution. Furthermore, an audit of new diagnosis and safe coprescribing was also conducted by our team across 31 of the 44 Hackney practices during the implementation phase, potentially causing contamination of control practices. However, the number of both trial intervention (n=13) and control (n=15) practices was comparable, suggesting equal exposure to this audit across both arms.[22] Although public health teams informally collaborate across Northeast London, crossover between boroughs is unlikely given that services are developed and budgeted separately. Our analysis will adjust for any potential contamination using various vertical and horizontal controls both within Hackney, as well as between boroughs using Newham, which has been devoid of any local service HIV promotional intervention, as control.

Moreover, the RHIVA2 trial and post-trial implementation will not be grouped as the same intervention. We will only use the post-trial implementation practices in the main analysis and only consider the trial intervention practices for the assessment of sustainability. Therefore, the sustainability analysis will have to be interpreted with caution with regards to the sustainability of the post-trial intervention, as the intervention in trial conditions might be different to that outside of trial conditions.

We may not be able to account for missing data such as from patients lost to follow-up after diagnosis in general practice, or from rapid testing pilots conducted in a small number of Newham practices during the study period.

Finally, the RHIVA2 trial intervention was adapted for the implementation phase (1) to expand promotion of routine offer of HIV testing from new registration checks to include sexual health and contraception appointments and (2) to enhance screening through additional use of venous blood sampling. This protocol adaptation was made to meet the needs of the local population through increasing access to testing, reflecting the complexities of translating a research intervention into clinical practice after completion of the research.

The results of the proposed analysis are likely to be of considerable interest to commissioners, particularly given that late diagnosis of HIV is a UK Public Health Outcomes Framework priority,[31] and that cost-effectiveness of HIV testing is complex, and becoming cost-effective in trial-based models only after a period of time has elapsed to counter the additional significant costs of early treatment.

**Author affiliations**
[1]Centre for Primary Care and Public Health, Queen Mary University of London, London, UK
[2]Population Health Research Institute, St George's, University of London, London, UK
[3]Department of Applied Health Research, University College London, London, UK
[4]Department of HIV and STI, National Infection Service, Public Health England, London, UK
[5]Barts Sexual Health Centre, Barts Health NHS Trust, London, UK
[6]Centre for Sexual Health, Homerton University Hospital NHS Foundation Trust, London, UK
[7]School of Health and Life Sciences, Glasgow Caledonian University, London, UK
[8]Specialised Commissioning Team, NHS England, London, UK

**Contributors** WL, LB, CN, ECB, SM, CE, JA, JF and CG significantly contributed to designing the study and drafted the protocol. FE-S, HM, KB, VD, AB, JH, VA, MS, SG, SC, MS and NF contributed to designing the study. All authors and contributors approved the submitted version of the manuscript.

**Funding** WL, FE-S, LB, CN, ECB were supported by the National Institute for Health Research (NIHR) Collaboration for Leadership in Applied Health Research and Care (CLAHRC) North Thames at Barts Health NHS Trust. HM was supported by an NIHR Doctoral Fellowship from 2013 to 2016. HM was supported by a National Institute of Health Research Doctoral Fellowship from 2013 to 2016.

**Disclaimer** The views expressed are those of the author(s) and not necessarily those of the NHS, the NIHR or the Department of Health.

**Competing interests** JA reports fees and non-financial support from Bristol Myers Squibb, grants and personal fees from Gilead Sciences, personal fees from ViiV, personal fees from Merck Sharp & Dohme, grants from Janssen, and personal fees from AbbVie, outside the submitted work. CE and JH report grants from Gilead Sciences.

**Ethics approval** Camden and Islington NHS Research Ethics Committee, London.

**Provenance and peer review** Not commissioned; externally peer reviewed.

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
