## [Reviewer comments · BMJ Open]

ARTICLE DETAILS

TITLE (PROVISIONAL)	Effectiveness and cost effectiveness of implementing HIV testing in primary care in east London: protocol for an interrupted time series analysis
AUTHORS	Leber, Werner; Beresford, Lee; Nightingale, C; Capelas Barbosa, Estela; Morris, Stephen; El-Shogri, Farah; McMullen, Heather; Boomla, Kambiz; Delpech, Valerie; Brown, Alison; Hutchinson, Jane; Apea, Vanessa; Symonds, Merle; Gilliam, Samantha; Creighton, S.; Shahmanesh, Maryan; Fulop, Naomi; Estcourt, Claudia; Anderson, Jane; Figueroa, Jose; Griffiths, Chris

VERSION 1 – REVIEW

REVIEWER	Ann Avery MetroHealth Medical Center Case Western Reserve University USA
REVIEW RETURNED	07-Jul-2017

GENERAL COMMENTS	There are numerous grammatical errors making it very confusing to read. Primarily, there are tense issues leaving the reader confused as to what has happened and what will happen but there are also numerous fragments and run-ons. Additionally, the information is very specific to London making it difficult for a non-local to understand and generalize the information. It is also difficult for the reader to not have defined the abbreviations the first time they are used. (though this may be what is recommended for the journal)
---

REVIEWER	Jan E.A.M van Bergen 1. STI AIDS Netherlands 2. University of Amsterdam, Department of General Practice, Academic Medical Centre, Amsterdam 1. No personal competing interest, no membership of Advisory committees, No fees. 2. I work for STI AIDS Netherlands. STI AIDS Netherlands is a non-profit organisation that receives funds from government and for a small part also from private parties such as Farma. There is a strict non-interference and transparency code. (https://www.soaids.nl/sites/default/files/documenten/Corporate/corporate_partnership_guidelines.pdf) 3. I am partner in the HTEAM initiative (www.HTEAM.nl), which is a multidisciplinary project working towards "The end of AIDS in Amsterdam". The HTEAM project receives funds from Aids fonds, Amsterdam Dinner, Janssen, VIIV, BMS Gilead, and the municipality of Amsterdam.
-----------------	---

GENERAL COMMENTS

I consider this paper as relevant and this research as very important as it addresses the crucial step from efficacy in trials towards (monitoring of) effectiveness in daily practice packed with all those multiproblem realities. Moreover the situation as described provides a unique opportunity given the specific setting with pre RCT, RCT and post RCT data, and a non-intervention borough.

I would suggest the authors to expand a bit more in their paper on the limitations. They present a lot of strengths of this proposal and stress the unique opportunity, but unfortunately this type of 'practical' implementation research also has many 'practical' problems that are hardly addressed in the paper. Eg the completeness (and biases) of reporting in the different boroughs; the cluster and cross-over effects of the interventions, the small differences between intervention and post-intervention implementation, and the way these factors might interfere (statistically) in the time series analysis.

The authors might consider to adjust their endpoint at page 10/28. Time between diagnoses and in-care is set at 2 weeks, 4 weeks, 3 months. However recent guidelines are trying to push this time-frame even more ahead (even within 1 week,.) It might be worthwhile also to display time between diagnoses and treatment as a plotter diagram to capture different outliers than those elected as indicators.

The authors should delete the double reference in the reference list (Ref 8 and 27 are similar.

VERSION 1 – AUTHOR RESPONSE

First of all, we would like to thank both reviewers for their very useful comments.

Response to Reviewer 1:

To improve clarity, we have corrected grammatical errors, we have expressed all planned study activity in future tense and defined the abbreviations when they first occur within the text.

We appreciate that the use of terms such as 'RHIVA2 intervention', 'RHIVA2 implementation', and the 'RHIVA2 programme' can be confusing to the reader. We have therefore clarified the differences between these terms by defining them at first use in the Introduction section and have used these terms consistently throughout the text.

To contextualise HIV testing policy in the UK, we have also added the following section to the second paragraph of the Introduction:

"In the Framework for Sexual Health Improvement in England (2013), the UK Department of Health additionally set out recommendations for the commissioning of HIV testing in high prevalence areas across all health care settings, including general practice.⁹ However, the decision to implement these recommendations lies with the 74 local authorities in England and the respective clinical commissioning groups (CCGs)."

Response to Reviewer 2:

We agree that implementation of research into clinical practice faces many practical issues. To better reflect the challenges of this implementation study, we have provided more detail on the differences between the RHIVA2 trial intervention and the post-trial implementation programme, as well as between the Hackney sexual health local enhance service (LES) and Tower Hamlets network improved service (NIS). We have also added several paragraphs on the potential limitations of our study, including the potential impact on the statistical analyses.

We have added the following additional study outcome measure:

- “The proportion of patients newly diagnosed with HIV in primary care that attend an HIV specialist department within 1 week of diagnosis.”

We believe the questions which will be addressed in this study should be based on the time between diagnosis and engagement with HIV services rather than initiation of treatment. Current HIV treatment policy in the UK is conservative and, although patients may be offered antiviral treatment at any level of CD4 count, firm treatment recommendations are only made to those with a CD4 count below or approaching 350. Therefore, time to start of treatment will be on an individual basis and average levels for this are available from national databases.

We have deleted the double reference as indicated above.